# Is There Reduced Hemodynamic Brain Activation in Multiple Sclerosis Even with Undisturbed Cognition?

**DOI:** 10.3390/ijms24010112

**Published:** 2022-12-21

**Authors:** Bianca Wagner, Clara L. Härig, Bertram Walter, Jens Sommer, Gebhard Sammer, Martin Berghoff

**Affiliations:** 1Department of Neurology, Justus-Liebig-University of Giessen, Klinikstrasse 33, 35385 Giessen, Germany; 2Bender Institute of Neuroimaging, Justus-Liebig-University of Giessen, Otto-Behaghel-Strasse 10H, 35394 Giessen, Germany; 3Department of Psychiatry, University of Marburg, Rudolf-Bultmann-Strasse 8, 35039 Marburg, Germany; 4Cognitive Neuro Science at the Centre of Psychiatry, Justus-Liebig-University of Giessen, Klinikstrasse 36, 35392 Giessen, Germany; 5Department of Psychology, Justus-Liebig-University of Giessen, Otto-Behaghel-Strasse 10F, 35394 Giessen, Germany

**Keywords:** multiple sclerosis, fMRI, cognitive impairment, attention network test

## Abstract

Cognitive impairments related to changes in deep gray matter and other brain regions occur in up to 70% of people with multiple sclerosis. But do such brain changes also occur in patients without significant cognitive impairment? Eighteen participants with relapsing-remitting multiple sclerosis (RRMS) and fifteen healthy controls participated in this study. Cognitive status, depression, and fatigue were assessed using the Multiple Sclerosis Inventory of Cognition (MUSIC), Beck’s Depression Inventory (BDI-II), and the Fatigue Severity Scale (FSS). fMRI was recorded while a participant performed the modified attention network test (ANT). The effects of ANT executive attention network on hemodynamic activation of a priori defined regions of interest, including the hippocampus, anterior cingulate cortex (ACC), thalamus, caudate nucleus, pallidum, and putamen were studied. The individual lesion load was estimated. For fMRI data analysis a general linear model with randomization statistics including threshold-free cluster enhancement as implemented in the FSL software was used. Participants with RRMS showed reduced activation of the executive attention network in the hippocampus, pallidum, and ACC. The thalamus was involved in both group activations but did not differ between groups. In summary, functional changes in the brain can also be demonstrated in RRMS patients without cognitive deficits. The affected brain regions can best be assigned to the attention network for executive control. This association could likely serve as a biological indicator of susceptibility to imminent cognitive impairment in MS.

## 1. Introduction

Multiple sclerosis (MS) is a chronic inflammatory and demyelinating disease of the central nervous system. Three main types of MS are differentiated, namely relapsing-remitting, primary progressive, and secondary progressive. Typical deficits associated with relapsing-remitting MS are vision problems, numbness, or tingling; progressive MS is characterized by spasticity, difficulties with walking and coordination, and gradual worsening of disability. Up to 70% of patients suffer from cognitive impairment, including deficits in attention, information processing, executive functioning, verbal fluency, or long-term memory [1,2]. Deficits in attention and information processing are most commonly reported [2,3]. Cognitive impairment may occur in the first months of the disease [4]. The prevalence of cognitive impairment increases with disease duration, and it is higher in patients with progressive MS [5]. Not surprisingly, cognitive impairment can affect a patient’s self-esteem, social functioning, ability to work, and quality of life [1,5,6]. It is therefore important to understand which CNS pathology is causing cognitive impairment in MS and to initiate early treatment. 

Deep gray matter (DGM) pathology has been associated with cognitive impairment in Lewy body dementia [7], Alzheimer’s disease [8], and MS [2]. Indeed, analyzes of autopsy tissue from patients with MS revealed demyelinated lesions and atrophy of various brain regions [9,10]. In addition, neuronal loss was noted in DGM lesions and non-demyelinated DGM regions [10]. Neuroimaging studies found atrophy of the subcortical DGM in patients with early-stage MS [11]. Atrophy of DGM nuclei (thalamus, caudate, putamen, pallidum) and of the hippocampus is correlated with cognitive impairment in patients with RRMS [2]. A significant correlation between cortical atrophy and poor performance in verbal memory, attention, and fluency has been reported in patients with RRMS and cognitive impairment [12]. Koenig et al., reported a decrease in hippocampal volume that correlated with decreases in episodic memory, attention, and processing speed (determined by the magnetization transfer ratio) [13]. Others have shown that T2 hypointensity of the subcortical gray matter, particularly in the caudate nucleus, pallidum, putamen and thalamus, is associated with cognitive impairment in patients [14]. In conclusion, these data suggest that degeneration of gray matter structures is associated with cognitive impairment in MS. 

Conversely, fMRI studies have shown that MS patients without cognitive impairment have more distributed and increased cerebral activation during cognitive tasks, as well as altered functional connectivity within so-called cognitive networks [15]. Cognition assessed by the Paced Auditory Serial Addition task revealed functional changes in MS in the right complementary motor area, the cingulate cortex, and bilaterally in the prefrontal, temporal, and parietal areas. These changes were associated with increased tissue damage, suggesting an adaptive mechanism in response to underlying neuronal disorganization or disinhibition [16]. Concerning tasks requiring sustained attention, information processing, and memory, patients with RRMS showed higher activation in subjects with better cognitive function than subjects with lower cognitive function. When neuropsychological testing is performed, patients with long-term MS and mild cognitive impairment showed increased and extensive activation of the frontal cortex and posterior parietal cortex. The effects decreased with task complexity and were most pronounced on the alertness task. The authors proposed a compensatory process through functional integration of frontal and parietal association areas [17].

Despite these studies on cognitive impairment in MS, the underlying early pathology of cognitive impairment in MS remains unclear. The aim of the present fMRI study was to investigate the activation of brain regions associated with cognitive impairment [2,13,14,16]. including the hippocampus, caudate nucleus, anterior cingulate cortex (ACC), thalamus, pallidum, and putamen in patients with RRMS. These regions are assigned to the executive networks of the brain. Therefore, in this study, this network was examined using Posner’s attention network model, with a focus on the executive attention network [18,19]. We decided to study patients with RRMS since this type is most common in young adults.

## 2. Results

### 2.1. Characteristics of Study Participants

Characteristics of the groups are shown in Table 1 and Table 2. In the patient group, seven participants had relevant symptoms of fatigue (FSS ≥ 4), two patients had mild depressive symptoms (BDI II score between 13–19 points), two patients showed mild cognitive dysfunction (MUSIC-score between 16–19 points; maximum score = 30). One patient performed poorly in the MUSIC-test due to a language barrier. Fourteen patients had low levels of vitamin D3 (25OHD3; normal range 16–74 ng/mL), one patient had a high level of TSH (normal range 0.4–2.5 mU/L). Two patients were excluded from the study due to moderate depression or severe cognitive deficits. In the control group, six participants had decreased levels of vitamin D3, and three controls showed increased levels of TSH. 

### 2.2. Prolonged Reaction Times in Patients 

To assess attentional processing, reaction times were recorded while performing the ANT. The effects of each attentional network are represented by reaction time differences between the respective conditions. The reaction time differences were (RRMS: HC; mean (sd)) 16.3 (32.7) ms: 38.3 (19.7) ms for the alerting network, 62.4 (39.4) ms: 86.4 (27.0) ms for the orienting network, and 116.2 (89.6) ms: 95.9 (30.4) ms for the executive network. These differences are in agreement with those found in other multiple sclerosis studies applying the ANT [20]. A mixed design ANOVA showed main effects for *Group* (F_2.68_ = 22.29, *p* < 0.001, η^2^_p_ = 0.4) but no interactions. In a mixed design ANOVA with the three main factors *Group* (RRMS vs. HC), *ExecAttent* (incongruent vs. not incongruent), and *Time block* (3 sections), all main factors showed significant effects. Response times were slower in patients than controls, in the incongruent condition compared to the other two conditions (congruent, neutral), and in Time block 1 (post hoc comparison *p* = 0.018). No interactions between these factors were found. Response time data are presented in Figure 1, the corresponding ANOVA table can be found in Table 3. 

### 2.3. Decreased Activation in DGM Structures in Patients

Figure 2 shows activation images thresholds using clusters determined by Z > 3.0 and a corrected cluster significance threshold of p = 0.05 for RRMS, HC and the mean of both groups.

*ExecAttent* effects were modeled by the GLM contrast HIT (incongruent) − (HIT (congruent) + HIT (neutral))/2. The TIME contrast included the reaction time regressor regardless of task type. In the patient group, activation was by *ExecAttent* in the thalamus (pFWE < 0.05). In the control group, regional activation by *ExecAttent* was also found in the thalamus, but also in the ACC, caudate nucleus, hippocampus, and pallidum (all pFWE < 0.05) (Table 4). Comparison of both groups revealed that activation by *ExecAttent* in patients was lower in the ACC, hippocampus and pallidum than in controls, but not in the thalamus (Table 5).

For the *Time* contrast, a regional activation associated with reaction speed was found in the ACC independent of the task type (pFWE = 0.017 at x/y/z = 6/20/32, cluster size of 31 voxels). However, no effects between the groups were observed above the threshold.

### 2.4. Lesion Load in the Brain

The destruction in the brain caused by MS was studied by measuring the lesion load (size and corresponding number of lesions) and the extent of white matter hyperintensities (corrected for estimated total intracranial volume) (Table 1). A strong correlation between both measures (rho = 0.97, *p* < 0.001) were found indicating a robust lesion estimation by the two algorithms. Lesion burden correlated with the EDSS (rho = 0.49, *p* = 0.026) but not with other tests such as MUSIC (rho = −0.15, *p* = 0.71), FSS (rho = −0.085, *p* = 0.62) and BDI II (rho = 0.013, *p* = 0.48).

## 3. Discussion

The most common and disabling cognitive deficits in patients are impaired attention, processing speed, and executive functioning. These deficits can occur early in the course of the disease [3]. The present study examined the hemodynamic activation of those brain areas, which have been associated with cognitive impairment in patients with relapsing-remitting multiple sclerosis (RRMS). Most of these areas belong to the executive attention network, which was examined by the attention network test (ANT) during fMRI. The effects of executive attentional effort (*ExecAttent*) and reaction speed (*Time*) on hemodynamic activation were analyzed. Other tests that have been applied recently to characterize deficits in those patients include the symbol digit modalities test (SDMT), the paced auditory serial addition test (PASAT), or the verbal learning and memory test (VLMT).

The main results of this study are less activation In the ACC, hippocampus, and pallidum in patients with RRMS. According to the description of attentional networks with the ANT by Fan et al. [21], this activation pattern indicates reduced executive control but sustained attention in participants with RRMS. Regarding executive attention (*ExecAttent*), activation of the ACC, caudate nucleus, hippocampus, pallidum, and thalamus was found in controls, while in patients only activation of the thalamus was observed.

Overall reaction speed was associated with activation of the anterior cingulate gyrus. No group differences were found suggesting that there were no major brain physiologic differences underlying behavioral indicators of general attentional function in patients or controls. However, the main result of the behavioral measures of attention was slower reaction times in patients. The behavioral results are in agreement with the work of Urbanek et al., describing prolonged reaction times in mostly cognitively unimpaired patients with RRMS by using the ANT [20]. In contrast to these studies, one study even reported a decrease in attentional performance over time [6]. The other effects, namely the increase of reaction times throughout the task and the time required to solve tasks of the incongruent condition, are well known and can be explained by a successful manipulation test. 

It has been suggested that atrophy of the hippocampus and DGM nuclei may be the best predictor of cognitive impairment [2]. DGM atrophy is already detectable in patients with early-stage MS [11]. In addition, patients have been shown to have reduced total thalamic volume as well as reduced neuron density [22]. However, the current study did not find changes in thalamic activation in cognitively normal patients, indicating an unimpaired attentional network associated with alertness. Without brain imaging, differences were found between patients with RRMS and controls in terms of alerting network behavioral measures [20]. The alertness network was not tested in the current study. 

Structural changes in the caudate nucleus, pallidum, and putamen were reported in patients with cognitive impairment suggesting that basal ganglia contribute to these deficits [14]. Increased cerebral activation, expanded cortical recruitment, and changes in the functional connectivity of regions associated with cognitive processing have been reported in MS patients [15]. This was explained by the authors as forced recruitment of cortical networks to compensate for cognitive deficits.

Cognitive processing is not limited to deep gray matter structures. A study of patients with RRMS found greater activation of the inferior and middle frontal gyrus, inferior parietal cortex, middle and superior temporal gyrus, anterior cingulate cortex, and basal ganglia. Hemodynamic activation was higher in patients with preserved cognitive function. Functional changes appeared to increase with greater tissue damage [16]. Studies in motor function have reported a positive correlation between changes in activation and the burden of cerebral lesions. Functional changes in cortical and subcortical motor areas may limit the disability caused by brain damage in MS [23].

There were several limitations in the study. The sample size is small, however. Post hoc power analysis using G*Power showed that a power (F-test, alpha = 0.05, total sample size = 33, two groups, 3 covariates, effect size 0.3) of Power (1-beta err prob) = 0.85 can be achieved. In addition, the age in the RRMS group was higher and the female/male ratio unequal. Further, long disease durations in two of the patients, associated with brain atrophy may have affected the results. However, testing for age effects provided no evidence for significant influence on the study results. The study focused on “executive attention” and associated brain areas, but interactions with other cognitive networks known to affects executive attention were not analyzed in the study.

In conclusion, less hemodynamic brain activation was found in the hippocampus, pallidum, and anterior cingulate cortex in patients with RRMS performing an executive attention network task. These anatomical regions can be assigned to the executive control network of attention. The achieved activation in the thalamus indicates an unimpaired attention network “Alertness”. The results of the study suggest an association between cognitive impairment and changes in functional integrity in gray matter, which could serve as a biological indicator of cognitive impairment in patients. These functional changes in the gray matter of the brain can also be detected without, or perhaps even before, cognitive deficits become noticeable.

## 4. Materials and Methods 

### 4.1. Study Participants

Twenty right-handed participants with RRMS were recruited from the outpatient clinic for multiple sclerosis and neuroimmunology at the Department of Neurology, Justus-Liebig-University, Giessen. These patients were diagnosed with RRMS according to the revised McDonald criteria [24]. A community sample (*n* = 15) was recruited as control group. For both groups, inclusion criteria were right-handedness, age between 18 and 60 years and an EDSS ≤ 4.5 [25]. Key exclusion criteria were moderate or marked cognitive dysfunction and moderate or severe depression. G*Power analysis [26] suggests, that a between-subjects ANCOVA with 33 participants across two groups would be sensitive to effects of partial η^2^ ≥ 0.28 with 80% power (alpha = 0.05).

The study was approved by the ethics committee of Justus-Liebig-University, Giessen (01/14). All participants gave their written informed consent. The experimental procedures were carried out in accordance with the Helsinki Declaration.

### 4.2. Study Procedures

All investigations were carried out in the Bender Institute of Neuroimaging, Justus-Liebig-University, Giessen, between 4 p.m. and 7 p.m. The Multiple Sclerosis Inventory of Cognition (MUSIC) [27] is a test for relevant cognitive deficits in MS. Language functions as well as general intellectual abilities are generally not affected. Thus, the test consists of 5 subtests for verbal learning and memory, verbal fluency, verbal and figural interference, and fatigue. Its sensitivity is 91%, its specificity is 85% [28]. The Beck Depression Inventory-II (BDI II) [29], and Fatigue Severity Scale (FSS) [30] were applied to all study participants. 

Complete blood cell count, thyroid hormones (TSH, T3, T4), and vitamin D3 levels were measured in all study subjects (this part of the study will be published elsewhere).

The Attention Network Test [19] (ANT) was used to probe cognitive load and attentional performance during functional imaging. The Attention Network Test (ANT) was developed to assess attention networks such as alertness, orienting and executive control [19]. The ANT has been applied to syndromes such as depression [31] and fatigue [32], in which patients often present with cognitive impairment. In patients with RRMS, the use of the ANT showed slower reaction times, indicating impairment of the alerting network [20]. In patients with progressive subtypes impairment of the orienting network was found [33]. In older subjects, cognitive fatigue, a frequent symptom in MS, was associated with executive attention but not with alertness or orientation [33]. Results concerning the association of DGM and other structures with the executive attention network are pending.

The function of the alerting network was assessed on the basis of reaction times (RT). The performance of the orientation network was estimated by RT modulated by cues indicating where the target will appear. Executive network function was evaluated by RT to the direction of a central arrow surrounded by flanks pointing to the same side, opposite side, or no direction. The effects on each attentional network are represented by reaction time differences between the respective conditions. The implementation of the task, including all timings, is shown in Figure 3. Participants were asked to pay attention to the center arrow on the display and respond quickly by pressing a button according to the direction indicated by the arrow. In each individual run, a fixation cross appeared in the center of the screen. Depending on the condition, a cue, which could be central, spatial, or absent, appeared, followed by the presentation of a fixation cross. After that, the target stimulus (middle arrow) was displayed along with the flanking stimuli (two arrows to the left or right of the middle arrow). The flankers could be aligned congruently or incongruently with the center arrow (target). With the participant’s response, the stimulus configuration was immediately replaced by the fixation circle. There was a maximum time window for a valid reaction, which was followed by a variable intertrial interval when switching to the next trial. The experiment consisted of 8 × 36 trials in which the 3 cue conditions × 2 target conditions were given in a fixed but balanced order such that each condition followed every other condition an equal number of times. For the current study, the ANT was modified by omitting the double cue condition. Fan et al. [18] found that double cue and center cue conditions are almost identical in terms of response times. In addition, Urbanek et al., reported an MS× congruent/incongruent × double cue/no cue interaction, suggesting that the results are at least ambiguous [20]. By reducing the number of tasks, we hoped to reduce the overall effects on participants’ fatigue.

During fMRI, ANT was presented using presentation version 17.2 (NeuroBehavioral Systems, Berkeley, CA, USA). Time for completion of the task varied between subjects; on average it was around 24 min.

### 4.3. MRI Acquisition

Brain images were acquired using a Siemens 3T prism scanner (Siemens, Erlangen, Germany) with a standard 64-channel head coil. First, a gradient echo field map sequence for B0 correction was measured with the following image geometry: echo time (TE) = 10.00 ms/12.46 ms; 90° flip angle; Duration = 3.44 min. For the functional data normalization procedure, structural images were acquired using a T1-weighted magnetization-prepared Rapid Acquisition Gradient Echo (MPRAGE) sequence: 176 sagittal slices; disk thickness = 0.94 mm; voxel size = 0.94 mm × 0.94 mm; repetition time (TR) = 1580 ms; inversion time (TI) = 900 ms; TE = 2.30 ms; Field of View (FOV) = 240 mm × 240 mm; 8° flip angle; acceleration factor = 3 (GRAPPA); and a T2-weighted (SPACE, dark liquid) sequence: 192 sagittal slices; slice thickness = 0.9; voxel size = 0.45 mm × 0.45 mm; TR = 5000 ms; TI = 1800 ms; TE = 387 ms; Field of View = 230 mm × 230 mm; 120° flip angle; Acceleration factor = 2 (GRAPPA). A T2*-weighted echoplanar imaging sequence (EPI) was used for fMRI: 40 slices; section thickness = 3.0 mm; distance factor = 25%; voxel size = 2.0 × 2.0 mm; TR = 2500 ms; acceleration factor = 2 (GRAPPA); TE = 30 ms; field of view = 220 mm × 220 mm; 85° flip angle; fat suppression; descending acquisition.

### 4.4. Statistical Analysis

#### 4.4.1. FMRI Analysis

Image preprocessing and analyzes were performed using the Functional Magnetic Resonance Imaging of the Brain (FMRIB) Software Library (FSL 5.0.9, Feat 6.00). The number of EPI volumes was dependent on the time for completion of the task, it was cut off 30 s after the last stimulation. Preprocessing included motion correction with McFLIRT 1, B_0_ unwarping, slice time correction with Fourier space-time series phase shifting, non-brain removal with BET [34], spatial smoothing with a 5 mm full-width half-maximal Gaussian kernel, and high-temporal filtering with a 100s cutoff. To detect outliers in functional scans, the minimum sum of squared differences from adjacent scans was calculated and thresholded using the method developed by Hubert and van der Veeken. [35] Functional images were registered to individual anatomical images using FLIRT by boundary-based registration [36,37]. Registration from high-resolution structural space to Montreal Neurological Institute standard brain space (MNI-152) was performed using a 12-parameter linear transformation (FLIRT) followed by a non-linear registration (FNIRT) [38,39]. Due to the failure of FNIRT in one subject, only a linear recording could be carried out for this participant.

#### 4.4.2. Functional Image Analyses

We hypothesized that higher executive attention demands associated with a slower reaction-time performance were indicated by increased hemodynamic brain activation. Therefore, for first-level analyses, the model consisted of the following regressors: for each condition (incongruent, congruent, neutral) regressors for correct responses, their first derivative, their modulation with the reaction times, regressors for wrong responses, and their first derivative. Additionally, the six motion parameters derived from the corresponding pre-processing step and a binary regressor for each outlier volume identified by the quality check pipeline before the pre-processing pipeline. In the basic work on ANT, Fan et al. [18] showed that neutral and congruent conditions differ neither in reaction times nor error rates. We choose to combine the congruent and neutral conditions and to contrast the incongruent condition. For this we used the term ExecutiveAttention (ExecAttent) instead of *Conflict*. Consequently, the contrast of interest was calculated as *ExecAttent* = HIT(incongruent) − [HIT(congruent) + HIT(neutral)]/2. This contrast was fed into the subsequent group-level analysis, which was performed according to the general linear model with pre-whitening [8]. Activation maps were calculated for the RRMS group, the control group, the mean of both groups and the difference between the groups (RRMS vs. HC). Statistical tests were performed for a priori defined regions of interest (ROI) [2,13,14,16]. These were the hippocampus, caudate nucleus, anterior cingulate gyrus, thalamus, pallidum, and putamen. ROI masks from the Harvard-Oxford Atlas included in the FSL 5.0.9 distribution. In order to assign a voxel to a specific structure, the membership probability had to be at least 0.5. To compare the groups, group-level design models were performed using permutation tests and a threshold-free cluster-enhancement statistic with familywise error correction in each ROI. The calculations were performed with the PALM software (Version alpha 102 running in Octave 3.8.1) [40] with tail approximation [41]. Group comparison was restricted to voxels that showed activation due to mental load in at least one group. Each ROI was considered separately.

#### 4.4.3. Response Times

Reaction times for each condition (congruent, incongruent, neutral) are provided as median for each subject. RT was analyzed using a mixed design ANOVA. Between-factor was *Group* (patients, controls), within factors were *Condition* (congruent, incongruent and neutral) and *Time*. *Time* refers to the 3 blocks of time, each block consisting of 96 trials introduced to account for the normally observed increase in reaction times over time. Where appropriate, Greenhouse-Geisser correction was calculated for the within-subject factors. Least significant differences (LSD) controlling for family-wise error were calculated for post-hoc comparisons [42]. IBM SPSS Version 26 was used computing the statistics. 

#### 4.4.4. Lesion Load Assessment

The Lesion Segmentation Tool (LST) Toolbox Version 2.0.15 [43] for SPM 12 [44] was used for individual lesion load assessment from the MPRAGE T1 structural images. The lesion growth algorithm (LGA) from the LST was started with an initial threshold of 0.3 (chosen after visual inspection) using T1- and T2-weighted structural images. T2-weighted images were not available for two subjects, therefore LGA could not be used. The corresponding values are missing in Table 1. In addition, the size of the white matter hyperintensities was calculated using Freesurfer version 6.0 [45,46]. For this purpose, the T1- and T2-weighted images were used (only the T1-weighted images in the two subjects with missing T2 images). The recon-all pipeline was invoked with the parameters ‘-mprage’, ‘-3T’. All segmentations and surface reconstructions were checked visually for errors. Statistics were calculated using PSPP (GNU PSPP, Version 1.2.0 https://www.gnu.org/software/pspp accessed on 20 December 2022).

## Figures and Tables

**Figure 1 ijms-24-00112-f001:**
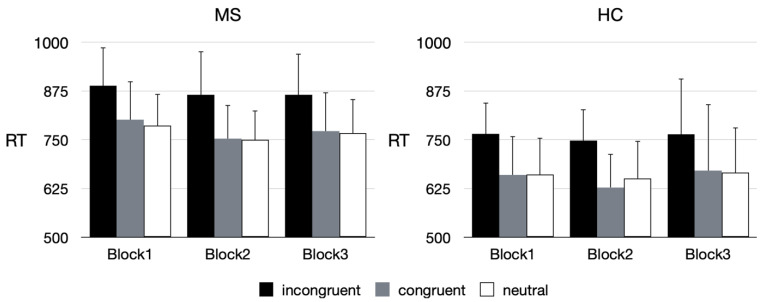
Reaction times for tasks, ExecAttent conditions, and Time blocks (Means + SD).

**Figure 2 ijms-24-00112-f002:**
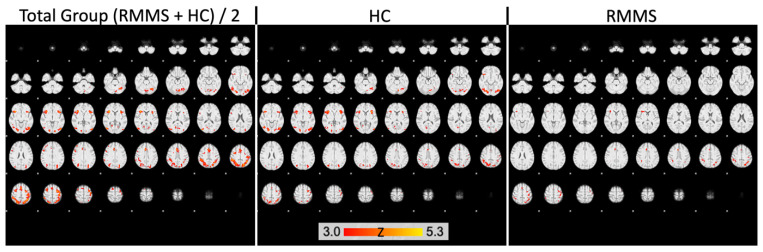
Activation images thresholds using clusters determined by Z > 3.0 and a corrected cluster significance threshold of p = 0.05. Images are shown for RRMS, HC and the mean of both groups.

**Figure 3 ijms-24-00112-f003:**
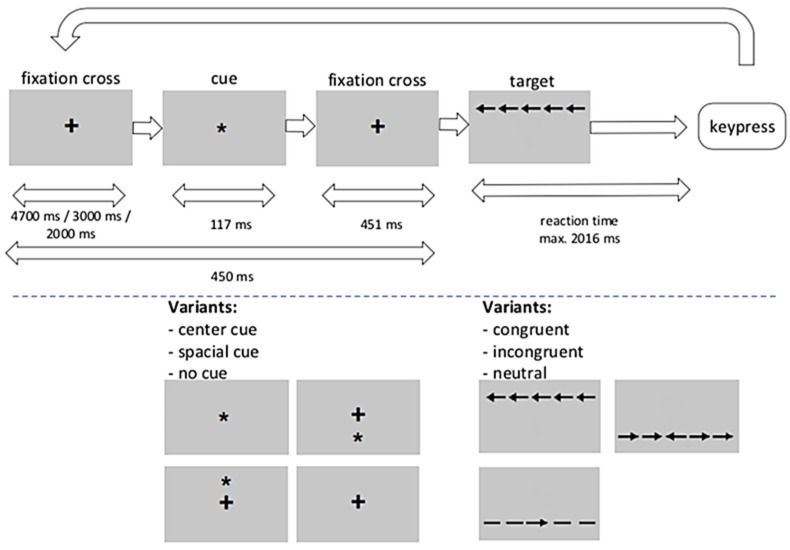
Modified ANT based on the version originally developed by Posner et al. [19]: upper panel, trial structure; lower panel, task variants. Symbols: + … fixation cross; * …spatial cue before target arrows and flankers appear; →, ← (middle) … displays the corresponding button for a correct answer; →, ← (other) flanking stimuli showing the same (congruent) or opposite (incongruent) side as the middle arrow, – … neutral flanking stimuli.

**Table 1 ijms-24-00112-t001:** Patient’s characteristics.

Patient	Sex (M/F)	Age (Years)	Years Since Diagnosis	EDSS (0–10)	FSS (1–7)	BDI II (0–63)	MUSIC (0-30)	LGA TLV	LGA n	WMH Volume	Disease Modifying Therapy
1	M	23	2	0.0	4.8	9	16	na	na	1250.9	Teriflunomide 14 mg
2	M	33	5	0.0	2.1	2	25	1.87	13	2185.2	IFNβ-1a 22 µg
3	F	54	24	1.5	1.1	0	27	5.03	20	3065.0	Teriflunomide 14 mg
4	M	37	6	0.0	1.8	0	28	0.22	6	699.7	IFNβ-1a 22 µg
5	F	51	8	1.0	1.9	2	23	na	na	1098.1	Teriflunomide 14 mg
6	F	33	2	1.5	5.9	11	29	0.20	5	566.0	IFNβ-1a 44 µg
7	F	55	25	3.0	5.0	19	25	1.16	19	1192.7	IFNβ-1a 44 µg
8	F	60	5	1.0	2.9	5	17	0.08	2	815.9	IFNβ-1a 22 µg
9	F	46	3	2.0	2.7	4	27	7.95	26	4741.1	IFNβ-1a 22 µg
10	F	36	3	2.5	6.4	17	25	0.26	5	543.4	IFNβ-1a 22 µg
11	F	54	11	2.5	5.2	0	26	2.82	24	1607.8	IFNβ-1b 250 µg
12	F	52	9	1.0	5.2	7	28	0.68	12	1202.2	IFNβ-1b 250 µg
13	M	49	10	4.5	4.6	10	23	5.31	24	4433.4	IFNβ-1b 250 µg
14	F	43	6	1.0	1.4	7	14 ^1^	1.02	17	1276.6	IFNβ-1a 22 µg
15	M	32	6	1.0	1.7	11	27	0.98	11	677.3	IFNβ-1a 44 µg
16	F	26	3	0.0	2.1	1	28	0.09	3	474.5	IFNβ-1a 22 µg
17	F	38	10	0.0	3.7	9	29	0.70	12	1387.1	IFNβ-1a 44 µg
18	F	33	10	1.0	1.9	0	24	0.15	4	666.4	IFNβ-1a 22 µg
Mean ± SD	5M; 13F	42.0 ± 11.0	8.1 ± 6.6	1.3 ± 1.2	3.4 ± 1.7	6.3 ± 5.8	24.5 ± 4.5				

¹ language barrier. na: not available, TLV: total lesion volume comparison of participants with RRMS (Table 1) and without (Table 2): Age (t = 3.2, df = 30.5, *p* = 0.003, d = 1.09), FSS (t = 2.4, df = 23.9, *p* = 0.026, d = 0.8), BDI II (t = 2.6, df = 22.3, *p* = 0.016, d = 0.9), MUSIC (t = −2.9, df = 27.4, *p* = 0.007, d = −1.0), Group difference from MUSIC controlled for age, depression, and fatigue (F = 2.29; *p* = 0.085).

**Table 2 ijms-24-00112-t002:** Healthy control characteristics.

Subject	Sex (M/F)	Age (years)	FSS (1–7)	BDI II (0–63)	MUSIC (0–30)
1	M	22	2.3	2	30
2	F	23	3.7	3	30
3	M	29	3.1	3	23
4	F	39	2.3	4	24
5	F	24	1.1	1	30
6	F	30	1.9	1	30
7	M	24	1.6	0	30
8	F	33	2.0	1	30
9	M	35	2.1	0	26
10	M	43	2.7	7	29
11	F	31	2.8	1	30
12	M	29	1.8	6	30
13	F	29	1.3	2	28
14	F	29	3.4	5	27
15	M	51	2.2	1	25
Mean ± SD	7M; 8F	31.4 ± 8.0	2.3 ± 0.7	2.5 ± 2.2	28.1 ± 2.5

**Table 3 ijms-24-00112-t003:** Reaction time analysis with factors Group, ExecAttent, and Time block: F, df (Greenhouse-Geisser corrected for within-subject effects), *p* and partial Eta². Non-integer df represent Greenhouse-Geisser adjusted degrees of freedom.

Effects	F	df	*p*	Partial η²
Group	12.20	1; 31	0.001	0.282
ExecAttent	211.63	1.82; 56.29	<0.001	0.872
Time block	5.00	1.47; 45.64	0.018	0.139
**Interactions**				
Group * ExecAttent	0.76	1.82; 56.29	0.460	0.024
Group * Time block	1.29	1.47; 45.64	0.277	0.040
ExecAttent * Time block	2.21	3.19; 98.76	0.087	0.067
Group * ExecAttent * Time block	0.76	3.19; 98.76	0.526	0.024

**Table 4 ijms-24-00112-t004:** Activation by ExecAttent in healthy controls (HC) and patients with RRMS (MS): Results of permutation tests using threshold-free cluster enhancement. Clusters containing only voxels with pFWE < 0.05 and their maximum with MNI coordinates are shown. Bonferroni threshold is pB = 0.05/6 = 0.008 considering tests for six ROIs. *p*-values ≤ pB are marked with an asterix.

Group Region	Cluster Size (Voxels)	p_FWE_	x	y	z
**HC**					
ACC	179	0.001 *	6	28	24
	201	0.008 *	2	12	32
Hippocampus	94	0.005 *	34	−18	−16
Pallidum	8	0.024	16	−4	−6
Caudate nucleus	20	0.009	12	6	12
Thalamus	255	0.003 *	−10	−8	10
	209	0.005 *	12	−12	6
Putamen	no result				
**RMMS**					
Thalamus	28	0.028	14	−10	2
other ROIs	no result				

**Table 5 ijms-24-00112-t005:** Group differences in activation by ExecAttent in healthy controls (HC) and patients with RRMS. Results of permutation tests using threshold-free cluster enhancement. Clusters containing only voxels with pFWE < 0.05 and their maximum with MNI coordinates are shown. Bonferroni threshold is pB = 0.05/6 = 0.008 considering tests for six ROIs. *p*-values ≤ pB are marked with an asterix. All tests indicate higher activation in HC than in MS.

Region	Cluster Size (Voxels)	p_FWE_	x	y	z
ACC	2	0.049	6	28	22
Hippocampus	55	0.002 *	32	−18	−16
Pallidum	8	0.003 *	16	−4	−6
other ROIs	no result				

## Data Availability

The data sets generated and/or analyzed in the course of the current study are not publicly accessible due to the applicable data protection law of the State of Hesse, but are available on justified request from the corresponding author.

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
