# Peer review of "Is There Reduced Hemodynamic Brain Activation in Multiple Sclerosis Even with Undisturbed Cognition?"

_ijms, 2022, doi:10.3390/ijms24010112_

Round 1

Reviewer 1 Report

Review of “Is there reduced hemodynamic brain activation in multiple

sclerosis even with undisturbed cognition?”

The manuscript investigated 18 RRMS patients and 15 HCs with MUSIC, BDI-II, and FSS. In addition, fMRI was performed testing ANT. HC showed fMRI activation in the ANT task in ACC, hippocampus, thalamus, and basal ganglia. RRMS patients showed reduced fMRI activation with increasing task complexity. Lesion load was associated with EDSS. Reaction time with ACC activation.

This is an interesting study on a highly important patient group. There are not many studies applying psychological testing, lesion mapping and task related fMRI in these patients.

However, results reported are not new and might be problematic in reproducibility because of small group sizes with rather inhomogeneous patients. In addition, the finding of decreased fMRI activation with increasing task complexity had been regularly reported to be modulated by compliance and is therefore not a specific biomarker for these patients. It had been described for several patient groups with brain damage but also elderly since decades. Astonishingly, the authors do not cite work with considerably larger patient groups using fMRI in attentional and cognitive velocity tasks for instance using the SDMT.

Positive:

- Clear writing

- appropriate fMRI measurement method

Problems in Detail:

- very low patient group size. There are huge cohorts also assessing cognitive velocity or attention network for RSMS available already.

- no emphasis on connectivity-based modifications in these patients. MS is a disease which is affecting the white matter primarily: so why not introducing these interaction deficits between the attention network right away? Why was white matter damage not quantified for the attention network relevant interactions?

- to much emphasis on the parts of the network but low general considerations that decreased compliance always results in a decrease in fMRI activation of a specific network tested.

- biomarkers not defined and specifically formulated here. There are suggestions for specific definitions in literature which should be followed

- no clear hypothesis on the modulation of the network tested with fMRI; no specific literature for each ROIs provided here. Why a statistical plan and power analysis has not been published before analysis as a trial (German Register or WHO registered trial)? This is a clinical study…

- very unspecific effects (reaction time); fMRI effects might be caused by everything (medication, depression, fatigue, lesion load, white matter damage, atrophy of GM)

- Table 4 and 5: provide t-values for highest activated voxels. Effects seem to be very small here- especially for between group comparisons

- fMRI evaluation and statistics: threshold-free cluster enhancement needs groups of ≥ 40 to provide reproducible data (Chen, Lu, Yan, HBM, 2018). The comparison effect between groups in ACC is based on 2 voxels and reached significance- strange…

- the Discussion is very short. Please include discussion of other studies using SDMT or cognitive tasks in RRMS. Please include limitations (see critical issues raised above).

- unimpaired thalamus activation; why only argue with alertness and not with central sensory input gait?

- a Figure of then lesion map might be provided.

Reviewer 2 Report

Thank you for the opportunity to review the paper! The study examined early signs in the brain for cognitive impairment in multiple sclerosis (MS) patients without significant cognitive impairment by measuring changes in the deep gray matter and other brain regions. Both MS patients and healthy controls completed the ANT while in the fMRI. They found the RRMS group showed reduced activation with increasing task complexity in the hippocampus, pallidum, and ACC compared to the control group. They concluded that functional changes in the brain can also be demonstrated in RRMS patients without cognitive deficits. The affected brain regions can best be assigned to the Attention Network for Executive Control.

The paper is well-written. This study is of high clinical relevance but hard to conduct with controlled sample characteristics that are important for the question tested. A major concern is that the analysis of the ANT was not based on specific attention network, making it hard to relate the RT results to fMRI findings; For example, although alerting didn’t seem to differ between the groups, how does it relate to behavioral measures of alerting? also given the interest of the paper is on task complexity, it is unclear to me how ANT indexes task complexity.

Additional comments:

Introduction.

1.       As the ANT measures 3 types of attention, alerting, orienting, and executive attention. However, the study only focuses on executive but not the others. Need to explain the rationale of this approach. Also, majority of the paper did not indicate specific types of attention but rather using attention broadly. This can be confusing for readers.

2.       Line 76. It’s unclear why the regions were chosen out of context.

3.       How do you define task complexity?

4.       Give an overview of MS types and why RRMS was chosen?

Methods.

1.       ANT task information such as # of trial, probability of each trial type should be reported in methods

2.        

Results.

1.       I might miss this analysis, but are the two groups age/gender-matched?

2.       Line 151, is there a missing word before sustained attention?

3.       Although overall RT was assessed for trial type, it might be good to report the RT measures for each attention type. It’d be interesting to see how two groups differ in each attention network

4.       Line 169-170. “…unimpaired attentional network associated with alertness.”. How about behavioral measures of alerting attention?

5.       Lesion load in the brain session, the correlation test results should be reported in addition to p.

6.       What’s the rationale of using this as task complexity? HIT (incongruent task) - (HIT (congruent task) + HIT (neutral task))/2.

7.       There are studies suggesting that using median rather than averaged RT of the ANT. The paper should justify the choice of using mean RT.

Discussion.

This session is overly simplistic, and some results were not fully discussed. for example, the current study reported “an unimpaired attention network "Alertness", but this contradicts to other studies did find impaired alertness. The paper should include the relevant literature and should at least consider alternative explanations.

Some factors should be mentioned in limitation:

1.       The MS group is not completely free of cognitive impairment and mental disorders (not excluded in healthy control either), which can be significant confounds in fMRI studies

2.       The age range is quite large in the MS group. What’s the role of age in brain activation during ANT? How did the analysis address this potential confound?

3.       the patients are in various stages of MS based on year since diagnosis, and atrophy of the subcortical DGM is already reported in patients with early-stage MS. How this impact the results is important to discuss.

Round 2

Reviewer 1 Report

Some of the concerns could be adressed but I have severe doubts on the validity of fMRI-results as reported here. In addition there are some typos left in the manuscript.

Abstract:

- Please insert the statistical method for fMRI-analysis applied here (threshold-free cluster-enhancement statistics) already in the abstract because the results obtained seem to be highly dependent on the statistical evaluation technique. 

Introduction:

-please insert references for the ROIs in the hypothesis in detail; the significance of effects is highly dependent on the ROIs selected – therefore the assumptions are crucial for this study.

- typo last sentence of the Introduction (your instead of young). 

- Methods: I´m not convinced that the fMRI statistics obtained here are reproducible. A significance with 2 voxels in a cluster when applying a cluster threshold is statistically problematic even in a ROI-approach. Height of activation is not provided- therefore a reviewer with more statistical expertise in the analysis technique used here should review that method in more detail. In addition, each ROI was considered separately (page 8, line 1) and not corrected for multiple comparisons.

- Saying that: it would be important to see the actual group maps for the fMRI-data to evaluate the method and its validity. Whey is that not provided here (data repository)?

- page 6: What does this sentence mean-typo in pace?

"Depending on the pace of work of the individual subjects, the total duration of the task was around 24 minutes.” 

Reviewer 2 Report

Thanks for submitting a revision.

Major concerns: If the focus is executive attention, then why not just use a flanker task instead of the ANT? This reflects poor experimental design and lack of research in the literature. Also, the paper needs to be carefully proofread. Several word usages can be improved. E..g, “most often in your adults.” most common is more appropriate. Line 162, inappropriate capitalized letter. Several acronyms need to be defined early on. This is a fMRI study but not showing any fMRI figures.

1.        It’d be good to add the characteristics of each after introducing the different types of MS

2.       Although only executive attention was assessed here, the cue conditions did not capture all four cue conditions in the original Posner’s ANT. This choice needs justification on why double cue condition was left out.

3.       There are still places using task complexity, e.g., line 96. Need to check the entire paper thoroughly

4.       “a priori selection of regions of interest is now explained based on the literature on cognitive impairment in MS.” Cannot find this information

5.       Lack of citation for critical information. E.g., line 101-104, line 143-145.

6.       Line 148-152. The description of calculating the attention networks by RT is unclear. How is it computed?

Results. The results sessions should be improved.

1.       ExNe is not a word and a confusing term. It needs to be spelled out fully when first used. It’s also not clear why it was defined as stated in line 212. Need to cite source.

2.       It’s not accurate to describe the test as repeated measures ANOVA as they are independent groups.

3.       Line 269-273. Lack of units for RTs. These results need to be discussed regarding how they fit into existing literature. If the alerting, orienting is not the focus of the paper, then the results can go into supplements so not to distract readers from the main point of the paper

4.       The ANOVA results were not reported properly in a publishable format. I’d encourage to put results in texts instead of listing in table 3. It’s also confusing to see the degree of freedom with decimals instead of integer.

5.       I had to read multiple time regarding the model of the ANOVA. Time was not listed as a main factor in line 229, but it looks like a factor later in 264. Need to clarify
